# Antibiotic use in companion animals in veterinary teaching hospitals in Thailand

**Sarin Suwanpakdee**[1,2], **Boonrat Chantong**[3], **Anuwat Wiratsudakul**[1,2], **Viroj Tangcharoensathien**[4], **Angkana Lekagul**[4], **Walasinee Sakcamduang**[1]*

**1** Department of Clinical Sciences and Public Health, Faculty of Veterinary Science, Mahidol University, Salaya, Phutthamonthon, Nakhon Pathom, Thailand, **2** The Monitoring and Surveillance Center for Zoonotic Diseases in Wildlife and Exotic Animals (MoZWE), Faculty of Veterinary Science, Mahidol University, Salaya, Phutthamonthon, Nakhon Pathom, Thailand, **3** Department of Pre-clinical and Applied Animal Science, Faculty of Veterinary Science, Mahidol University, Salaya, Phutthamonthon, Nakhon Pathom, Thailand, **4** International Health Policy Program Foundation, Nonthaburi, Thailand

* walasinee.sak@mahidol.ac.th

## Abstract

The high volume of antibiotics used for companion animals, off-label use of human-registered antibiotics for veterinary purposes, and close human-animal interactions raises policy concerns related to antimicrobial resistance in companion animals. This study aimed to assess the volume and type of antibiotic usage in companion animals at veterinary teaching hospitals in Thailand. In 2018, we obtained a dataset of visits that occurred between 2015 and 2017 from eight selected veterinary teaching hospitals with a high caseload of companion animals. In total, we included 938,522 dogs and 242,342 cats in our study. Total antibiotic consumption was estimated, and antibiotic usage was standardized by calculating the amount of antibiotic consumed by weight per year (mg/kg/year). The top five antibiotics used include amoxicillin–clavulanic acid, enrofloxacin, cephalexin, doxycycline, and metronidazole, all of which are commonly used to treat bacterial and parasitic infections. Critically important antimicrobials for human medicine, such as amoxicillin–clavulanic acid and enrofloxacin, were among the most used. From 2015 to 2017, total antibiotic usage (kg) markedly decreased (i.e., by 57.0%), with a particularly notable reduction of 78.2% observed for amoxicillin–clavulanic acid. Moreover, veterinarians' overall prescriptions of antibiotics registered for human use decreased by 16% over the same period. However, there was a notable increase in the use of injectable antibiotics registered for human use relative to oral forms. Furthermore, we observed a relatively high usage of third-generation cephalosporins, which may contribute to the emergence of antimicrobial resistance in companion animals. This study emphasizes the need to educate veterinarians and veterinary students on the rational use of antibiotics and highlights the importance of regular monitoring and surveillance of antibiotic use in companion animals.

**Data availability statement:** All relevant data are within the manuscript.

**Funding:** This study was funded by the Food and Agriculture Organization of the United Nations (OSRO/RAS/502/USA) and implemented by the International Health Policy Program Foundation. The funders had no role in study design, data collection and analysis, decision to publish, or preparation of the manuscript.

**Competing interests:** The authors have declared that no competing interests exist.

## Introduction

Widespread and frequent use of antibiotics in humans and animals has contributed to antimicrobial resistance among both pathogens and commensal bacteria. In 2018, the World Health Organization (WHO) recognized antimicrobial resistance (AMR) as an emerging crisis [1]. A previous study estimated that AMR may cause an annual mortality count of 10 million worldwide by 2050 [2]. Companion animals are often considered to be members of the family. The close contact between pets and owners can increase the risk of transmitting antimicrobial-resistant bacteria and genes between humans and pets. Given this close relationship, companion animals are often treated with extensive use of antibiotics for bacterial infections [3].

Some antibiotics have been targeted toward veterinary medicine; however, most are also directly applied to human regimens [4]. For example, most South African veterinarians prescribe off-label use of human-registered antibiotics, and pet owners can also provide human antibiotics to treat their pets [5]. In Thailand, most antibiotics for humans and animals can legally be purchased from retail pharmacies without prescription [6]. Thus, antibiotic self-medication is not uncommon [7,8].

Dogs are the most common pets raised in Thailand. The human-to-owned dog ratio in Thailand was estimated at 6.4, based on an owned dog population of approximately 11.2 million [9]. Moreover, veterinary teaching hospitals serve as premier institutions for clinical training, knowledge dissemination, and research innovation, thereby contributing to advancing veterinary science and improving society [10]. As of 2018, there were 13 veterinary teaching hospitals affiliated with 10 universities in Thailand. Veterinary teaching hospitals in Thailand share many of the same core functions and characteristics as those in other countries, but they also reflect the unique context of Thai veterinary education, animal health needs, and cultural attitudes toward animals [11–13]. Furthermore, hospitals have a complete prescription record of antibiotic use as well as the number of dog and cat visits. However, these hospitals are responsible for the prevention, diagnosis, and treatment of common to complex veterinary clinical conditions.

To effectively address AMR, it is crucial to understand antibiotic consumption patterns in all sectors, including companion animals, which have been neglected when crafting policy responses to AMR. A prior study in the UK found that antibiotics were prescribed to one-fourth of all dogs visiting veterinary hospitals [14]. Despite the availability of annual reports on antibiotic consumption in livestock, particularly swine, poultry, and other food animals [15], to date no studies have investigated the volume of antibiotics consumed by companion animals in Thailand.

One of the goals of Thailand's national strategic plan on AMR (2017–2021) is to achieve a 30% reduction in animal antibiotic consumption by 2021 [15]. Subsequently, there have been various studies in Thailand that have documented total animal consumption of animals. However, these indicate that total consumption of swine, poultry, and other food animals is very large, chiefly because Thailand is a major exporter of food animal products [16]. A greater understanding of the volume, profile, and off-label use of human antibiotics in companion animals in the context of the emergence of resistant pathogens will contribute to effective policies and interventions in the companion animal subsectors.

This study estimated the volume of antibiotics, including the WHO List of Critically Important Antimicrobials (CIA) for Human Medicine [17], drug preparations, and types of drug registration (i.e., for humans or animals) used by small animals in selected veterinary teaching hospitals.

## Materials and methods

### Study design

In 2018, we conducted a retrospective study to gather comprehensive data on antibiotic usage in companion animals (i.e., cats and dogs) in eight veterinary teaching hospitals in Thailand. This study included both outpatient and inpatient cases from each of the hospitals. Overall, these institutions are veterinary teaching hospitals that provide comprehensive veterinary services and operate primarily as tertiary care centers. The hospitals also function as referral centers and offer a wide range of diagnostic and therapeutic services, ensuring consistency and comparability in the level of care across all study sites. The study covered the period from 2015 to 2017 and focused on hospitals with high caseloads of companion animals. Our main objective was to quantify antibiotic usage and identify patterns of antibiotic use. We focused primarily on the types of antibiotics used and their relevance to public health concerns related to AMR.

### Ethical approval

The Animal Care and Use Committee of the Faculty of Veterinary Science, Mahidol University, Thailand, granted a formal waiver of ethics approval due to the retrospective nature of the study. We obtained permission from all selected veterinary teaching hospitals to engage in data sharing. Our research did not require human research ethics approval following the researcher-assessment guide for human research (see: https://sp.mahidol.ac.th/eng/assessment.html).

### Data collection

Data on antibiotic usage were obtained from annual inventory records that document the antibiotic purchases of each participating veterinary hospital. This inventory data was recorded electronically as part of the hospital's routine management systems. Furthermore, we invited hospital administrators to participate via an official letter and obtained their permission to access aggregated antibiotic inventory data. All participating hospitals provided aggregated information, including the quantity of antibiotics purchased, the number of companion animals treated, and the proportion of cases that were treated as opposed to those receiving preventative treatment (e.g., vaccinations and routine health check-ups).

To ensure consistency in data collection, all veterinary teaching hospitals applied a standardized summary form that included the total number of animal visits, the total amount of each type of antibiotic used per year, preparation type (i.e., oral, topical, or injectable), the types of drug registration (i.e., for humans or animals), and antibiotic categories following the criteria specific in the WHO List of CIA for Human Medicine [17]. This form ensured consistent data collection across all hospitals and provided a comprehensive overview of antibiotic usage. Moreover, since the study was not able to determine accurate animal weight, an average of 4.1 kg for cats and 19.1 kg for dogs [18] was used for estimation purposes.

### Data management and statistical analyses

Data entry and descriptive analysis were performed using Microsoft Excel (version 2016). The complete dataset was then validated by checking values for consistency, plausibility, and coherence. Incorrect values were treated as "missing."

In this study, the total amount of each antibiotic used in the year $i$ was estimated by multiplying the number of drug containers purchased by each hospital in that year by the corresponding drug strength, as shown by the following formula.

$$T_{ABO_i} = \sum\nolimits_{j=1}^{m} Number\ of\ drug\ container_j \times Drug\ strength_j$$

(1)

Here, *i* = 1, 2, 3, …, n: refers to the specific year of data collection,

 *j* = 1, 2, 3, …, m: refers to a drug container of a specific antibiotic

In 2015, we encountered a lack of data on antibiotic use from two out of eight veterinary teaching hospitals. To address this gap, we estimated the total antibiotic use for these two hospitals by using the median value of antibiotic consumption recorded in 2016 and 2017. Following this, we calculated the overall antibiotic usage from 2015 to 2017. To evaluate our estimates, we conducted a sensitivity analysis comparing the existing data on total antibiotic use across 2015, 2016, and 2017 with our estimated data for the two hospitals that were missing information in 2015. This analysis aimed to determine the difference in total antibiotic use between the two estimation methods and assess whether we could rely on the existing data without the estimated information from those two hospitals.

All animals that were presented for vaccination were excluded. The use of antibiotics was analyzed and presented in the antibiotic amount per kilogram animal weight per year (mg/kg/year). The following equation was used to estimate the antibiotics used (mg) in an animal per kilogram weight per specific year *i* ($A_i$).

$$A_i = \frac{T_{ABO_i}}{(N_{Di} \times W_D) + (N_{Ci} \times W_C)}$$

(2)

$N_{Di}$= is the total annual number of dogs in a specific year *i*; $N_{Ci}$ is the total annual number of cats in a specific year *i*; $W_D$ is the average dog weight (kg); and $W_C$ is the average cat weight (kg). For the average body dog and cat weights we estimated using standard weights of 19.1 and 4.1 kg, respectively [18].

The top five antibiotics have been identified based on their ranking in terms of total grams of antibiotic usage, as well as the measurement of antibiotic consumption expressed in milligrams per kilogram per year.

Additionally, we monitored antibiotic usage from 2015 to 2017 and categorized the proportion of those antibiotics that were included on the WHO List of Critically Important Antimicrobials (CIA) for Human Medicine (version 2019). The classification included three categories: 1) Critically Important Antimicrobials (CIA), 2) Highly Important Antimicrobials (HIA), and 3) Important Antimicrobials (IA) [17].

## Results

### Sample profiles and total antibiotic use

Of the 13 veterinary teaching hospitals invited, eight (61.5%) agreed to participate in the study. The size of these teaching hospitals varies by staff level and service load. On average, 24 (2–81) full-time veterinarians provided services to 53,676 (3,382–210,017) animals each year. We primarily counted the numbers of dogs and cats, since other species were present only in very small numbers. A total of 1,180,864 animals, which included both animals seeking treatment and those not seeking treatment, were presented to these hospitals between January 2015 and December 2017. Of these animals, 938,522 (79.5%) were dogs and 242,342 (20.5%) were cats. Furthermore, after combining numbers from all eight hospitals, the total number of animals seeking treatment was 1,105,814 (i.e., 363,120 in 2015, 375,976 in 2016, and 366,718 in 2017) (Table 1). The sensitivity analysis revealed that the total antibiotic usage, after adjusting for missing data, changed by only 1.18% compared to the original dataset, excluding the unreported data from 2015 (S1 Table). Consequently, we decided to use the original data instead of the estimated values for the two hospitals when illustrating total antibiotic use from 2015 to 2017, as shown in Fig 1.

### Usage of antibiotics on the WHO CIA for Human Medicine List

The total antibiotic consumption in cats and dogs, measured by active ingredients, between 2015 and 2017 was 1,391.4 kg, comprising 726.3 kg in 2015, 352.4 kg in 2016, and 312.7 kg in 2017. The proportion of antibiotics prescribed to animals in eight veterinary teaching hospitals ranged from 35.0% to 84.3% of all animal visits. Antibiotic consumption was reduced by 57.0% from 2015 to 2017 (Fig 1 and S2 Table).

**Table 1. Number of cats and dogs seeking treatment in eight veterinary teaching hospitals (2015–17).**

| Hospital | Number of dogs | | | | Number of cats | | | |
|---|---|---|---|---|---|---|---|---|
| | 2015 | 2016 | 2017 | Total | 2015 | 2016 | 2017 | Total |
| A | 153,953 (52.6%) | 156,767 (52.3%) | 149,208 (52.1%) | 459,927 (52.4%) | 42,299 (59.8%) | 44,325 (58.1%) | 45,487 (56.5%) | 132,111 (58.1%) |
| B | 18,154 (6.2%) | 17,929 (6.0%) | 18,412 (6.4%) | 54,495 (6.2%) | 3,207 (4.5%) | 3,851 (5.0%) | 4,490 (5.6%) | 11,548 (5.1%) |
| C | 41,570 (14.2%) | 40,223 (13.4%) | 36,800 (12.9%) | 118,593 (13.5%) | 11,859 (16.8%) | 11,112 (14.6%) | 11,376 (14.1%) | 13,840 (6.1%) |
| D | NA | 2,425 (0.8%) | 2,286 (0.8%) | 4,711 (0.5%) | NA | 1,216 (1.6%) | 1,320 (1.6%) | 2,536 (1.1%) |
| E | 23,792 (8.1%) | 23,977 (8.0%) | 22,873 (8.0%) | 70,642 (8.0%) | 5,754 (8.1%) | 6,025 (7.9%) | 7,271 (9%) | 19,050 (8.4%) |
| F | 3,700 (1.3%) | 4,147 (1.4%) | 2,947 (1.0%) | 10,794 (1.2%) | 0 (0.0%) | 0 (0.0%) | 0 (0.0%) | 0 (0.0%) |
| G | NA | 4,943 (1.6%) | 7,138 (2.5%) | 12,081 (1.4%) | NA | 2,440 (3.2%) | 3,681 (4.6%) | 6,121 (2.7%) |
| H | 51,241 (17.5%) | 49,294 (16.4%) | 46,535 (16.3%) | 147,069 (16.7%) | 7,591 (10.7%) | 7,302 (9.6%) | 6,894 (8.6%) | 21,788 (9.6%) |
| **Total** | **292,410** | **299,705** | **286,199** | **878,314** | **70,710** | **76,271** | **80,519** | **227,500** |

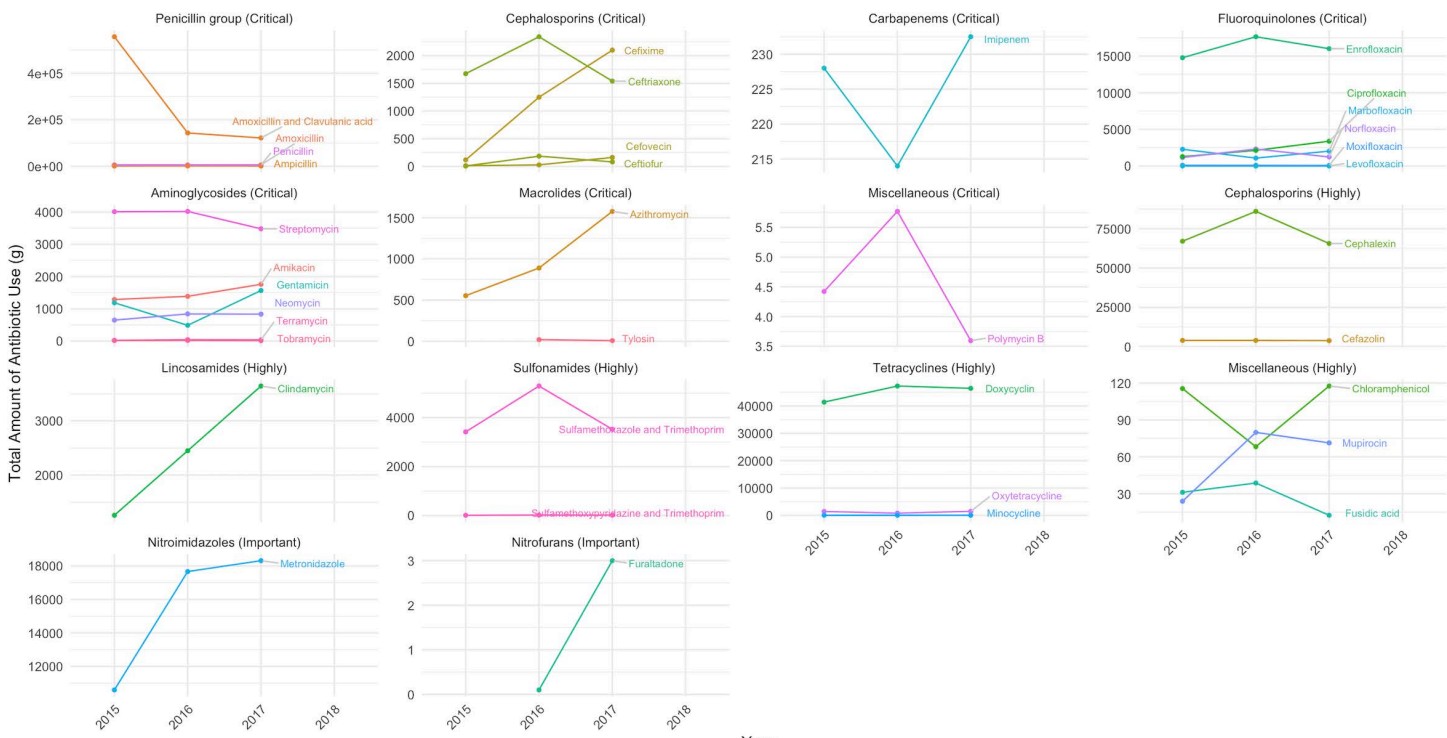

**Fig 1. Total amount of antibiotics used between 2015 and 2017.** Amounts are shown in grams of active ingredient for all substances on the WHO CIA List for Human Medicine.

Of the total 1,391.4 kg of antibiotics used, 955.6 kg (68.7%) comprised CIA, 398.2 kg (28.0%) included highly important antimicrobials, and 46.6 kg (3.4%) consisted of important antimicrobials (Fig 1). Amoxicillin–clavulanic acid (821.8 kg) and enrofloxacin (48.5 kg) were prescribed in the highest total amounts (in kg) of all CIA-listed treatments. Within the category of highly important antimicrobials, cephalexin (218.7 kg) and doxycycline (135.1 kg) had the highest total amount prescribed. In addition, metronidazole was an important antimicrobial with the highest total amount prescribed at 46.6 kg.

Next, we analyzed the usage of the five most commonly administered antibiotic agents—referred to here as the top five antibiotics—ranked by the total milligrams of active substance administered per kilogram of dog and cat body weight per

year, during the period from 2015 to 2017. This analysis revealed a substantial reduction in the use of amoxicillin–clavulanic acid, which decreased from 94.8 mg/kg/year in 2015 to 23.7 and 21.0 mg/kg/year in 2016 and 2017, respectively. In contrast, the use of doxycycline, metronidazole, cephalexin, and enrofloxacin fluctuated among the three years (Table 2). Additional details on antibiotic use per kilogram of animal body weight per day between 2015 and 2017, categorized by veterinary teaching hospital, are provided in S3 Table.

The top five antibiotics used in companion animals between 2015 and 2017 included amoxicillin–clavulanic acid, cephalexin, doxycycline, metronidazole, and enrofloxacin. Moreover, the total amount of amoxicillin–clavulanic acid prescribed by veterinarians was very high in 2015, but its use substantially decreased between 2016 and 2017 (Fig 2).

## Antibiotic use by administration route

There are three administrative routes: injection, oral administration, and topical administration. Twenty-two groups of antibiotics were administered via injection, with amoxicillin–clavulanic acid, penicillin, enrofloxacin, and streptomycin being the most frequently used CIA substances (66.4%), whereas only cefazolin was the highly important antimicrobial substance prescribed in substantial amounts. Twenty antibiotics were administered orally. Of these, amoxicillin–clavulanic acid, cephalexin, doxycycline, enrofloxacin, and metronidazole were the top five most prescribed oral antibiotics by total grams. Sixteen antibiotics were administered topically, and of these, amikacin was the most prescribed (70.0%) (Fig 3).

Data on antibiotic consumption from the eight veterinary teaching hospitals revealed that there were notable variations among different administration routes between 2015 and 2017 (Fig 3). Overall, injectable antibiotics showed a stable pattern over the three years, with stable usage rates for penicillin and streptomycin. Oral antibiotic use showed varying patterns depending on the individual substance, with some increasing and others decreasing over time. For example, amoxicillin–clavulanic acid was highly prescribed in 2015 but not in 2016 and 2017. Meanwhile, cephalexin, doxycycline, metronidazole, and enrofloxacin all showed increased consumption. With respect to topical antibiotics, amikacin showed the highest consumption over the three-year period, and its use slightly increased in 2017. Neomycin was consistently used overall years, while mupirocin was increasingly used in 2016 and 2017. When considering all routes combined, we note that the overall usage of antibiotics decreased. However, amoxicillin–clavulanic acid was consistently one of the most used antibiotics over the three years. Finally, the top five antibiotics used by animal weight (mg/kg/year) were in line with the top five total antibiotics used (in total grams) (Table 2 and Fig 3).

## Off-label antibiotic use

An analysis of antimicrobial use by registration type (i.e., whether the antimicrobial was intended for human or animal use) revealed variations among the three administration routes between 2015 and 2017. Among the injectable antibiotics, 71.1% were specifically designated for veterinary use in 2015. However, this proportion decreased to 62.5% in 2016 and 38.1% in 2017. This means the use of human-registered injectable antibiotics in small animals has steadily increased, from 28.9% in 2015 to 37.5% in 2016 and 61.9% in 2017.

In 2015, only 7.7% of oral antibiotics administered to animals were specifically registered for veterinary use, while the majority (92.3%) were antibiotics designated for human use. This pattern shifted in subsequent years, with a notable increase in the use of animal-registered oral antibiotics to 25.3% in 2016 and 23.0% in 2017. Moreover, the use of human-registered oral antibiotics in animals decreased from 92.3% in 2015 to 74.7% in 2016 and 71.1% in 2017. With respect to topical antibiotics, 100% of those used were antibiotics designated for human use.

Thus, although most antibiotics used in small animals over all administrative routes are registered for human use, their use showed a slight increased from 2015 to 2017. (Fig 4).

**Table 2. Annual amount of antibiotic substance (mg/kg) administered to dogs and cats from 2015–2017.**

| Antibiotic group | Generic name | Amount of antibiotic used (mg/kg/year) | | |
|---|---|---|---|---|
| | | 2015 | 2016 | 2017 |
| **Critically important antimicrobials** | | | | |
| Penicillin | Penicillin | 0.9 | 0.9 | 0.9 |
| | Ampicillin | 0.2 | 0.2 | 0.2 |
| | Amoxicillin | 0.7 | 0.7 | 1.0 |
| | Amoxicillin and Clavulanic acid | 94.8 | 23.7 | 21.0 |
| Cephalosporin | Cefovecin | $0.2 \times 10^{-2}$ | $0.5 \times 10^{-2}$ | $3.0 \times 10^{-2}$ |
| | Ceftiofur | $0.1 \times 10^{-2}$ | $3 \times 10^{-2}$ | $1.0 \times 10^{-2}$ |
| | Ceftriaxone | 0.3 | 0.4 | 0.3 |
| | Cefixime | $2.0 \times 10^{-2}$ | 0.2 | 0.4 |
| Carbapenem | Imipenem | $4.0 \times 10^{-2}$ | $4.0 \times 10^{-2}$ | $4.0 \times 10^{-2}$ |
| Fluoroquinolone | Enrofloxacin | 2.5 | 2.9 | 2.8 |
| | Moxifloxacin | $2.0 \times 10^{-2}$ | $2.0 \times 10^{-2}$ | $1.0 \times 10^{-2}$ |
| | Norfloxacin | 0.2 | 0.4 | 0.2 |
| | Marbofloxacin | 0.4 | 0.2 | 0.4 |
| | Levofloxacin | $0.6 \times 10^{-3}$ | $0.6 \times 10^{-3}$ | $0.4 \times 10^{-3}$ |
| | Ciprofloxacin | 0.2 | 0.4 | 0.6 |
| Aminoglycoside | Amikacin | 0.2 | 0.2 | 0.3 |
| | Gentamicin | 0.2 | 0.1 | 0.3 |
| | Neomycin | 0.1 | 0.1 | 0.1 |
| | Tobramycin | $0.3 \times 10^{-2}$ | $0.3 \times 10^{-2}$ | $0.2 \times 10^{-2}$ |
| | Streptomycin | 0.7 | 0.7 | 0.6 |
| | Terramycin | $0.3 \times 10^{-2}$ | $0.7 \times 10^{-2}$ | $0.7 \times 10^{-2}$ |
| Macrolide | Azithromycin | 0.1 | 0.2 | 0.3 |
| | Tylosin | NA | $0.3 \times 10^{-2}$ | $0.1 \times 10^{-2}$ |
| Miscellaneous | Polymycin B | $0.7 \times 10^{-3}$ | $1.0 \times 10^{-2}$ | $0.6 \times 10^{-3}$ |
| **Highly important antimicrobials** | | | | |
| Cephalosporin | Cephalexin | 11.4 | 14.3 | 11.3 |
| | Cefazolin | 0.7 | 0.6 | 0.6 |
| Lincosamide | Clindamycin | 0.2 | 0.4 | 0.6 |
| Sulfonamide | Sulfamethoxazole and Trimethoprim | 0.6 | 0.9 | 0.6 |
| | Sulfamethoxypyridazine and Trimethoprim | $0.1 \times 10^{-2}$ | $0.2 \times 10^{-2}$ | $0.2 \times 10^{-2}$ |
| Tetracycline | Oxytetracycline | 0.2 | 0.1 | 0.3 |
| | Doxycycline | 7.1 | 7.8 | 8.0 |
| | Minocycline | $0.3 \times 10^{-2}$ | $0.4 \times 10^{-2}$ | $0.7 \times 10^{-2}$ |
| Miscellaneous | Chloramphenicol | $2.0 \times 10^{-2}$ | $1.0 \times 10^{-2}$ | $2.0 \times 10^{-2}$ |
| | Fusidic acid | $1.0 \times 10^{-2}$ | $1.0 \times 10^{-2}$ | $0.2 \times 10^{-2}$ |
| | Mupirocin | $0.4 \times 10^{-2}$ | $0.1 \times 10^{-2}$ | $0.1 \times 10^{-2}$ |
| **Important antimicrobials** | | | | |
| Nitroimidazole | Metronidazole | 1.8 | 2.9 | 3.2 |
| Nitrofuran | Furaltadone | NA | $0.1 \times 10^{-4}$ | $0.5 \times 10^{-3}$ |

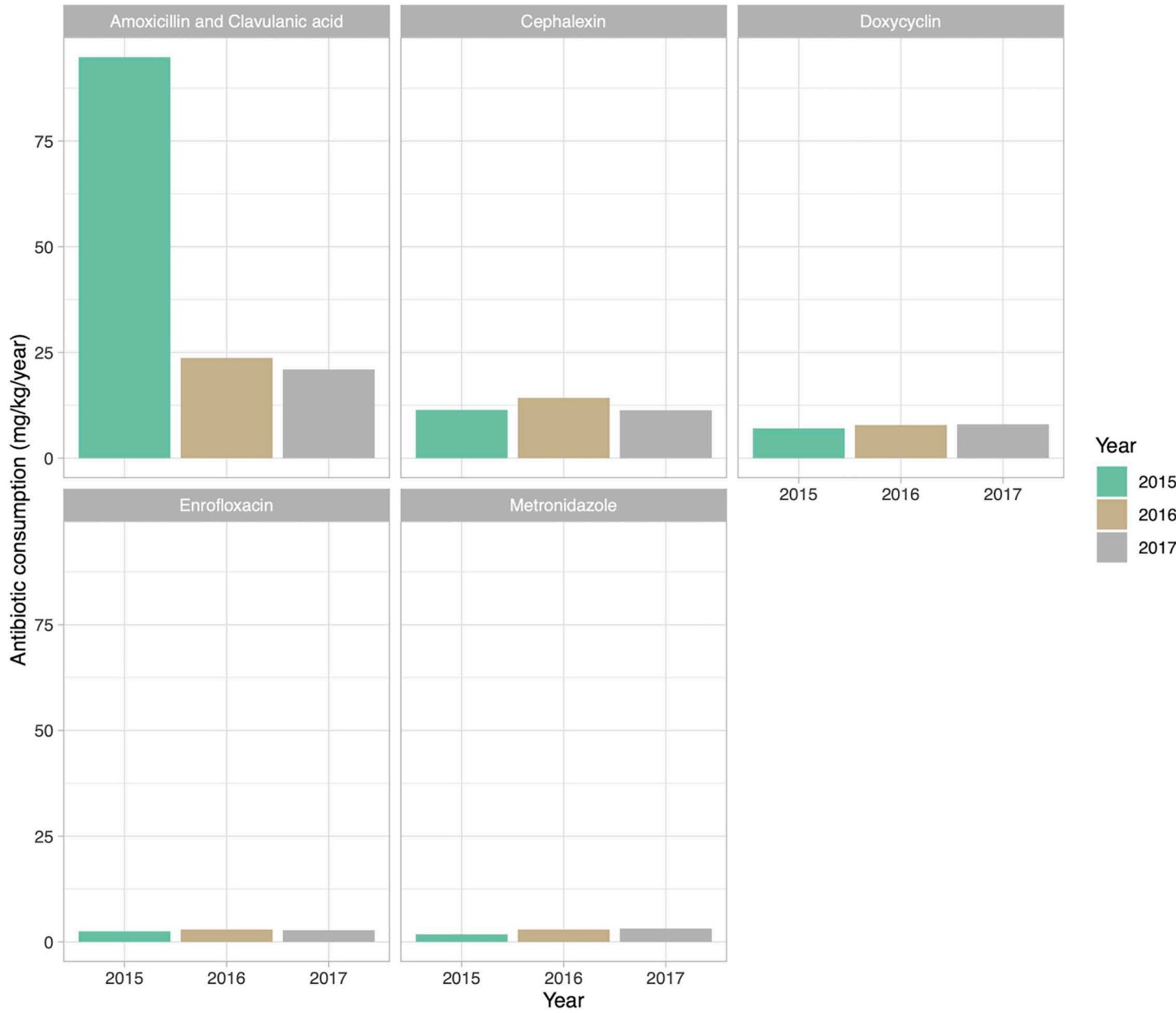

**Fig 2. Top five antibiotics used in cats and dogs between 2015 and 2017, ranked by total milligrams per kilogram of animal weight per year.**

## Discussion

To our best knowledge, this is one of the first studies conducted in a middle-income Southeast Asian country to esti-mate antibiotic use in companion animals, especially in veterinary teaching hospitals that maintain electronic records of antibiotic use. Our results show that of the 1,180,864 animals examined, 1,105,813 (93.6%) were given antibiotics between 2015 and 2017. Next, we estimated the usage of the top five most highly prescribed antibiotic substances. These amounts, estimated by total amount (mg), were similar to the top five most highly prescribed antibiotics as measured in

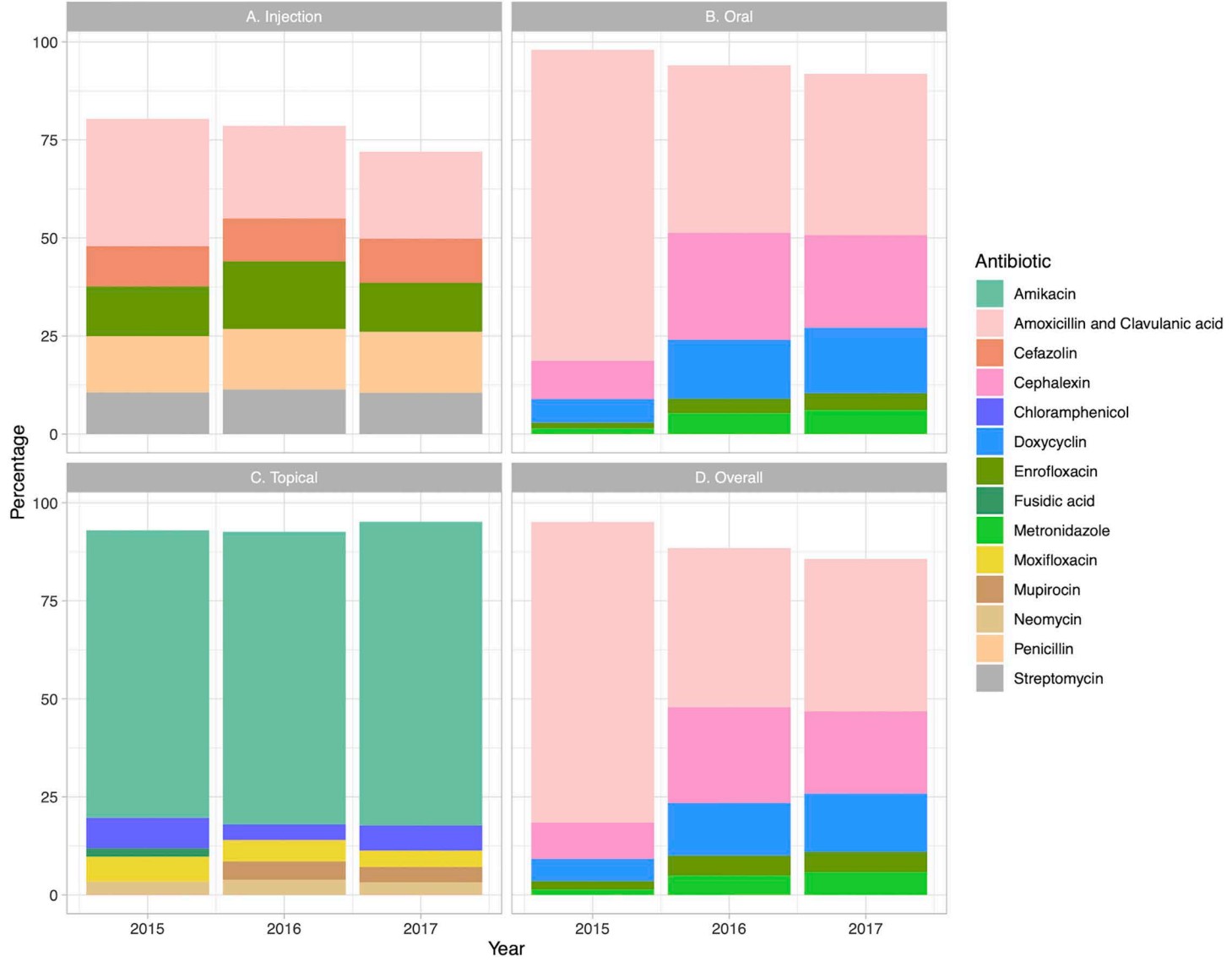

**Fig 3. Administration routes of the top five antibiotics (in grams) prescribed annually from 2015 to 2017.**

mg/kg. We note that our study was not able to collect the weight of each animal. Thus, a standardized weight 4.1 kg for cat and 19.1 kg for dog was used instead [18].

The high rates of antibiotic prescriptions (35.0%–84.2%) observed in cats and dogs at veterinary teaching hospitals in Thailand highlight a notable reliance on antimicrobial therapy for treating clinical conditions. The relationship between antimicrobial use (AMU) and AMR in companion animals is complex. However, studies of livestock have reported a positive association between the quantity of antimicrobials used and the prevalence of AMR [19,20]. However, findings from another study suggested that other factors—i.e., beside the high volume of antibiotics used—such as inappropriate usage, may be important contributors the development of AMR in companion animals [21].

Although there is no electronic prescription database used by these hospitals, the form-based data sharing protocol used here provides valuable insight into antibiotic usage patterns over three years. Moreover, our data also reports

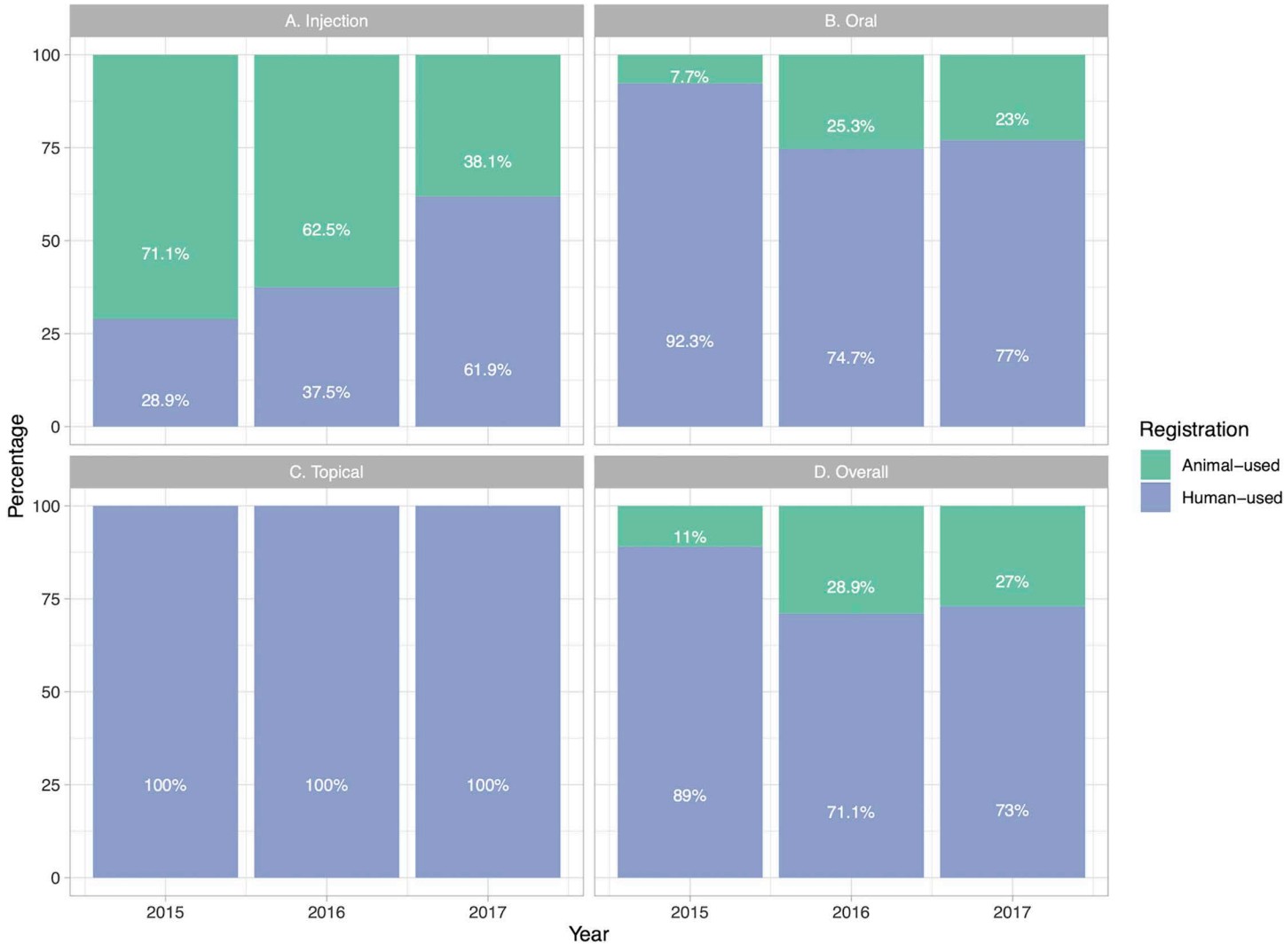

**Fig 4. Antibiotic use in small animals by substance registration status.**

classification data, especially regarding WHO CIA group, type of registration, and administrative route, all of which can inform policy design to rectify challenges associated with antibiotic usage. We note that other researchers have estimated antibiotic usage by obtaining data via direct retrieval from medical records or from prescription databases in veterinary hospitals and clinics [22,23]. Other recent studies have evaluated antibiotic use in companion animals in veterinary teaching hospitals, including in, the USA, France, Italy, and Germany [13,24–26].

The top three most prescribed antibiotics (i.e., amoxicillin–clavulanic acid, cephalexin, and doxycycline) identified by our study were consistent with reports from veterinary teaching hospitals in the UK [19], France, Italy, and Germany [13,24,26]. In contrast, oxytetracycline and metronidazole were found to be predominantly used to treat blood parasites and acute gastroenteritis in Nigeria [27]. Finally, a study of 14 veterinary teaching facilities in the US found that potentiated penicillin was most commonly used for systemic treatment of both cats and dogs [24].

The WHO CIA lists, prioritizes, and reserves the most critical antimicrobials used in human medicine. Our findings confirm the widespread and excessive use of CIA-listed antimicrobials, including amoxicillin–clavulanic acid. Moreover,

the observed reduction in the use of CIA-listed drugs from 2015 to 2017 may be due to several reasons. First, the reduction may be caused by increased awareness among veterinarians, in response to various educational campaigns, regarding the rational use of antimicrobials [6]. Second, the interpretation of our findings should also consider the concurrent introduction of major antimicrobial stewardship initiatives. For example, Thailand's National Strategic Plan on AMR was launched in 2016 [28], and specific Veterinary Council guidelines for companion animal antibiotic use followed in 2017. Our study, which covers the period 2015–2017 captures this pivotal transition, and therefore provides a valuable baseline that reflects the lead-up to as well as the initial implementation phase. We recognize explicitly that having only one full year pre- (2015) and one year coinciding with the launch/start of these initiatives (2017) limits our ability to definitively assess their sustained impact. However, these data effectively highlight usage patterns within veterinary teaching hospitals right at the point these crucial guidelines and strategies were introduced.

The use of CIA for human medicines in small animals is alarming; here we found that 68.7%, 28.0%, and 3.3% of all antibiotics used in cats and dogs belonged to the critically important, highly important, and important antimicrobial categories, respectively (Fig 1). Moreover, these findings were consistent with other studies [21,29,30]. Annual antibiotic use by animal weight showed that amoxicillin–clavulanic acid was the most frequently prescribed CIA for cats and dogs (Fig 2), a finding that is also consistent with other studies [21,29,31]. Amoxicillin–clavulanic acid is a broad-spectrum antibacterial agent that is frequently used in both human and veterinary medicine. Cephalexin was the second most used antibiotic, followed by doxycycline, enrofloxacin, and metronidazole. In some years, enrofloxacin and metronidazole were either the fourth or fifth most used drugs. In general, potentiated amoxicillin and cephalexin are used to treat skin, respiratory tract, and urinary tract infections in cats and dogs. Prior studies have found that the use of cephalexin is linked to the emergence of methicillin-resistant coagulase-positive staphylococci in dogs, which can be transmitted to humans [32,33]. Furthermore, another study found that *Escherichia coli* obtained from dogs exhibited complete resistance to amoxicillin–clavulanic acid [34].

Our analysis also found relatively high usage of third-generation cephalosporins (e.g., cefovecin, ceftiofur, ceftriaxone, and cefixime) in veterinary teaching hospitals. These drugs are classified as "CIA for human medicine" by the WHO, and the use of third-generation cephalosporins in veterinary medicine has been found to increase the emergence of AMR in various regions. For instance, one study in Japan found that the use of third-generation cephalosporins in veterinary medicine was associated with the development of extended-spectrum cephalosporin resistance in *E. coli* infecting dairy cattle [35]. Another study in Brazil demonstrated that dogs treated with ceftriaxone or ceftiofur had a higher occurrence of extended-spectrum cephalosporin-resistant Enterobacterales [36]. Furthermore, a study from Denmark revealed that frequent use of third- and fourth-generation cephalosporins was linked to the existence of extended-spectrum cephalosporin-resistant *E. coli* in pig herds [37].

Next, we noted that the observed variation in antibiotic consumption across different administration routes suggests that several factors, including changes in prescription practices, differences in clinical presentations, and evolving bacterial resistance profiles can influence these patterns. In general, the broad-spectrum efficacy of amoxicillin–clavulanic acid and cephalexin make them favorable choices for a variety of infections. The use of doxycycline in 2016 and 2017 may be attributed to changes in clinical guidelines and the availability of alternative therapies during the period of analysis. Moreover, fluctuating in the use of injectable antibiotics such as imipenem and ceftriaxone may be explained by the frequency of serious infections that require systemic effects [38]. Moreover, the consistent use of penicillin and streptomycin indicates their continued relevance for specific infections that require injectable formulations. Amikacin is the most commonly used topical antibiotic because it is broad spectrum and can be used to treat a wide variety of bacterial infections. It is also an active ingredient in various commercially accessible preparations, including mupirocin and neomycin. These topicals can be effective for certain skin infections with limited systemic side effects. Finally, we suspect that variation in moxifloxacin use for eye infections such as conjunctivitis may be explained by variation in prevalence [39],

  

The veterinary teaching hospitals in this study were aware of AMR and introduced antimicrobial stewardship initiatives aimed to realize optimal use of antibiotics and to strengthen rational use practices. This intervention likely contributed to the observed overall reduction in antibiotic use. Furthermore, this campaign may also motivate the high use of broad-spectrum antibiotics, especially amoxicillin–clavulanic acid, since it is highly effective in treating many clinical conditions in companion animals. However, there remains a need to explore its replacement by antibiotics in the important antimicrobial group that may be equally effective.

Variation in administration routes reflect a tailored approach to different infections, and may align with best practices for minimizing resistance and maximizing therapeutic outcomes. The continued monitoring of AMR and antibiotic use should guide effective antimicrobial stewardship, and are therefore essential for addressing the dynamic challenges of AMR in small animals.

Antibiotic crossover use involves using antibiotic formulations that are licensed for human use in animals. As depicted in Fig 3, a decline in the proportion of animal-registered injectable antibiotics from 71.1% to 61.9% over the three-year period indicates that human-registered formulations are still needed for managing systemic infections, particularly in severe animal infections. The market availability of numerous generic antibiotics that are registered for human use may account for the high usage rates observed in small animals in this study. The existence of low-cost, high-quality generic versions of human medications likely makes them more affordable to pet owners who pay treatment costs.

In contrast, the marked increase in the use of animal-registered oral antibiotics (i.e., from 7.7% to 23.0%) reflects market availability and an overall preference for veterinary-specific oral formulations. Further, the "food flavor" applied to oral formulations supports high adherence by pets. Oral forms that foster palatability, different dosage forms, and pharmacokinetics that better suit animal physiology support the use of antibiotics registered for animal use. Thus, the observed reduction in the use of human-registered oral antibiotics from 92.3% to 77.0% suggests that there is an ongoing effort to limit the use of human antibiotics for treating animal disease. Improved compliance with veterinary practices can minimize the risk of AMR emergence due to cross-species differences in drug metabolism and efficacy. Moreover, the availability and stability of topical antimicrobials in humans can help explain the common use of these medicines in pets. Therefore, improved availability and lower cost of generic versions of certain preparations, including injectable and topical medications, whether drugs are registered for human use, drug preparation flavor, and the existence of palatable oral formulations are all factors that may explain variation in antibiotic use in small animals.

This study revealed that all human-registered injectable antibiotics (e.g., amikacin, clindamycin, imipenem, ampicillin, cefazolin, cefovecin, ceftriaxone, and penicillin G) as well as all oral antibiotics registered for human use (e.g., furazolidone, metronidazole, azithromycin, ciprofloxacin, norfloxacin, sulfamethoxazole/trimethoprim, and cefixime) are also used by companion animals. Moreover, the market availability of lower-cost generic antibiotics registered for human use explains this important finding.

Remarkably, two of eight veterinary teaching hospitals reported using imipenem, a broad-spectrum carbapenem antibiotic that is one of the CIA (and which should be strictly reserved for severe human infections) during 2015–2017. One hospital used imipenem for all three years, and one started using it in 2017 (S2 Table). This practice raises concerns regarding the increased prevalence of AMR in humans [40] as well as the high mortality associated with AMR [41]. Practices such as the use of imipenem should be discouraged and replaced by the use of other potent antibiotics.

Policies to restrict the use of imipenem in veterinary teaching hospitals are supported by well-documented clinical evidence. First, the broad spectrum of activity can introduce selective pressure and cause the emergence of AMR, especially in multidrug-resistant organisms. Veterinary practitioners should prescribe imipenem only if absolutely necessary according to antibiotic sensitivity tests. Second, the high cost of imipenem relative to other antibiotics may be a deterrent for pet owners if other equally effective antibiotics are available. Third, drugs like imipenem, a member of a critically important antibiotic group, should be reserved for use only as a last-resort antibiotic for human use; it should therefore be used judiciously in veterinary settings [38]. In this study, we identified one hospital that had consistently used imipenem for

 

three years. Introducing interventions designed to strengthen antimicrobial stewardship may help limit use of imipenem to infections in which other antibiotics are not effective.

Next, the observed shift in antimicrobial use from reliance on those registered for human use toward increased use of animal-registered antibiotics is a positive change, since it indicates increasing levels of species-appropriate treatments. This supports optimization of therapeutic outcomes in veterinary practice and can minimize future development of AMR.

However, the drivers of antimicrobial use extend beyond individual choice and policy implementation. Indeed, a complex interplay of social, economic, cultural, and historical determinants markedly influences antimicrobial use patterns within a population. For instance, ethnographic studies have observed the normalization of antimicrobial use as a perceived solution for both human health concerns and agricultural production demands in contemporary society. Consequently, addressing AMR effectively will necessitate systemic interventions that transcend focusing only on modification of individual behavior [42]. Furthermore, a previous study has underscored the importance of this multifaceted approach, advocating for consideration of the broader socio-political-economic context in which antimicrobial use occurs. Therefore, context-specific solutions, beyond simplistic awareness campaigns, are crucial for tackling AMR [43]. Nevertheless, in the Thai veterinary sector, studies that have integrated social science methodologies into studies of AMR remain limited. Further investigations employing such interdisciplinary approaches are strongly encouraged.

Though this study improves our understanding of antibiotic use in companion animals, especially regarding critically important antibiotics and the use of medications designated for human use, several limitations are also present. (1) These data were collected retrospectively in 2018 and cover the period from 2015 to 2017. While these findings provide important baseline information, they may not reflect current antibiotic use practices following recent updates in AMR policy and veterinary guidelines. (2) Since our data were collected only from veterinary teaching hospitals, which often differ in caseload complexity and available resources relative to private veterinary clinics, these findings may not be generalizable to all veterinary settings in Thailand. (3) The lack of individual animal weight data required the use of standardized average weights (i.e., 4.1 kg for cats and 19.1 kg for dogs) derived from a previous study in the Netherlands. These figures may overestimate the average weight of Thai animals and may thereby potentially lead to an underestimation of antibiotic use per kilogram. (4) This study did not capture the specific clinical conditions or indications for which antibiotics were prescribed. As a result, it was not possible to evaluate whether prescriptions were aligned with evidence-based or guideline-recommended practices. (5) Not all veterinary teaching hospitals maintained comprehensive data regarding antibiotic use, and this may have an impact on the reliability of our findings. (6) One limitation of our study was the absence of antibiotic usage data from two hospitals in 2015. However, we addressed this by estimating the missing values using the median antibiotic usage figures of these hospitals in 2016 and 2017 [44], reflecting a plausible usage pattern approach. A subsequent sensitivity analysis showed that total antibiotic usage, after imputation, differed by only 1.18% from the original dataset, excluding the missing 2015 data (S1 Table). This minimal difference suggests that the available data remain a valid and representative estimate of the overall antibiotic usage across all eight hospitals over the 2015–2017 period. (7) Differences in the affordability of veterinary services in veterinary teaching hospitals and private clinics can influence the use of antibiotics registered for animal or human use. Despite these limitations, this study contributes to the growing body of evidence reporting on antimicrobial use in companion animals. Moreover, it underscores the importance of continued surveillance, targeted stewardship interventions, and improved access to veterinary-specific antibiotic formulations for promoting the rational use of antibiotics.

The present study also possesses several notable strengths. It is among the first to assess antibiotic usage in companion animals across multiple veterinary teaching hospitals in a middle-income Southeast Asian country. Furthermore, by including data from eight teaching hospitals, which collectively represent 60% of such institutions in Thailand, this study provides a broad overview of prescription patterns in high-caseload clinical settings. Furthermore, this study offers valuable insight into the use of CIAs and to what extent off-label use of human-registered antibiotics occurs in small animal-focused practices—an area that has been largely underexplored in this region.

## Conclusions

This study highlights a notable reduction in antibiotic use in companion animals between 2015 and 2017 across eight veterinary teaching hospitals in Thailand. This change may demonstrate the effectiveness of existing strategies aimed at combating AMR. Despite these positive outcomes, the continued use of CIA and human-registered antibiotics in veterinary practices underscores the need for strengthened antibiotic stewardship. Enhancing professional education on courses related to AMR and the rational use of antibiotics among in-service veterinarians is recommended. Furthermore, policies should be revised to restrict the use of critically important antibiotics for human medicine, and should encourage the adoption of alternative drug classes where appropriate.

Use of antibiotics in companion animals has historically been overlooked by both policy frameworks and Thailand's national action plan on AMR. Given the close contact between pets and humans, companion animals may act as reservoirs for AMR bacteria that can easily transfer to humans, thus highlighting the importance of surveillance in this sector. Consequently, establishing robust monitoring systems for AMR and antibiotic use in companion animals is strongly recommended to address this critical gap.

Future studies should broaden the scope of this study by including a more diverse range of veterinary hospitals and should aim to more accurately characterize antibiotic use in companion animals in Thailand. Finally, expanding the study to include species beyond cats and dogs may provide a more comprehensive understanding of antibiotic use in companion animals.

## Supporting information

**S1 Table. Estimated impact of missing data imputation on total antibiotic use in dogs and cats, 2015–2017.**
(XLSX)

**S2 Table. Total amount of antibiotics used, in milligrams of active ingredient, in the WHO List of CIA for Human Medicine between 2015 and 2017.**
(XLSX)

**S3 Table. Antibiotic use in animals (mg/kg/day) between 2015 and 2017.**
(XLSX)

## Acknowledgments

We thank the veterinary teaching hospitals for providing information regarding antibiotic use and our colleagues at Mahidol University for assistance with data cleaning.

## Author contributions

**Conceptualization:** Sarin Suwanpakdee, Boonrat Chantong, Anuwat Wiratsudakul, Walasinee Sakcamduang.

**Data curation:** Sarin Suwanpakdee, Boonrat Chantong, Anuwat Wiratsudakul, Walasinee Sakcamduang.

**Formal analysis:** Sarin Suwanpakdee, Boonrat Chantong, Anuwat Wiratsudakul, Walasinee Sakcamduang.

**Funding acquisition:** Boonrat Chantong, Viroj Tangcharoensathien, Angkana Lekagul, Walasinee Sakcamduang.

**Investigation:** Sarin Suwanpakdee, Boonrat Chantong, Anuwat Wiratsudakul, Walasinee Sakcamduang.

**Methodology:** Sarin Suwanpakdee, Boonrat Chantong, Anuwat Wiratsudakul, Walasinee Sakcamduang.

**Project administration:** Boonrat Chantong, Walasinee Sakcamduang.

**Resources:** Walasinee Sakcamduang.

**Software:** Sarin Suwanpakdee, Anuwat Wiratsudakul.

**Supervision:** Viroj Tangcharoensathien, Walasinee Sakcamduang.

**Validation:** Sarin Suwanpakdee, Boonrat Chantong, Anuwat Wiratsudakul, Walasinee Sakcamduang.

**Visualization:** Sarin Suwanpakdee, Boonrat Chantong, Anuwat Wiratsudakul, Angkana Lekagul, Walasinee Sakcamduang.

**Writing – original draft:** Sarin Suwanpakdee, Boonrat Chantong, Anuwat Wiratsudakul, Walasinee Sakcamduang.

**Writing – review & editing:** Sarin Suwanpakdee, Boonrat Chantong, Anuwat Wiratsudakul, Viroj Tangcharoensathien, Angkana Lekagul, Walasinee Sakcamduang.

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
