## [Decision Letter · Decision Letter 0]

28 Mar 2025

PONE-D-24-54379Antibiotic use in companion animals in veterinary teaching hospitals in ThailandPLOS ONE

Dear Dr. Sakcamduang,

Thank you for submitting your manuscript to PLOS ONE. After careful consideration, we feel that it has merit but does not fully meet PLOS ONE’s publication criteria as it currently stands. Therefore, we invite you to submit a revised version of the manuscript that addresses the points raised during the review process.

We look forward to receiving your revised manuscript.

Kind regards,

Md. Tanvir Rahman, DVM, MSc (Canada), PhD (UK), FBAS

Academic Editor

PLOS ONE

Journal Requirements:

2**. ** Thank you for stating the following financial disclosure:

“This study was funded by the Food and Agriculture Organization of the United Nations (OSRO/RAS/502/USA) and implemented by the International Health Policy Program Foundation.”

“We thank the veterinary teaching hospitals for providing information on antibiotic use and our colleagues at Mahidol University for assisting with data cleansing. We also acknowledge the Food and Agriculture Organization of the United Nations and the International Health Policy Foundation, Thailand, for their funding support and valuable suggestions.”

“This study was funded by the Food and Agriculture Organization of the United Nations (OSRO/RAS/502/USA) and implemented by the International Health Policy Program Foundation.”

**Additional Editor Comments:**

Please revise the manuscript as suggested by the reviewers.

Reviewers' comments:

Reviewer's Responses to Questions

**Comments to the Author**

1. Is the manuscript technically sound, and do the data support the conclusions?

Reviewer #1: Yes

Reviewer #2: Partly

2. Has the statistical analysis been performed appropriately and rigorously? 

Reviewer #1: Yes

Reviewer #2: N/A

3. Have the authors made all data underlying the findings in their manuscript fully available?

Reviewer #1: Yes

Reviewer #2: Yes

4. Is the manuscript presented in an intelligible fashion and written in standard English?

Reviewer #1: No

Reviewer #2: Yes

5. Review Comments to the Author

Reviewer #1: Dear Author

I suggest to revise the manuscript for the English because in some portion it is difficult do understand.

I also suggest to verify the references in the text because some are not perfectly connected to the sentence and there are some newer references you can add for example for the antimicrobial use in veterinary hospitals

I would suggest to expired the antimicrobials in table 2 in grams instead of mg

Reviewer #2: Summary:

Antibiotic usage in companion animals is little researched compared to livestock, particularly outside of high-income countries. This manuscript therefore provides novel insight by investigating the nature and type of antibiotics used to treat cats and dogs in Thailand. To do this, the authors descriptively analyse data collected from veterinary teaching hospitals. Whilst this is an important knowledge gap, I have concerns about the methods used and the claims made about longitudinal trends.

Abstract

I suggest the authors reconsider the results they present in the abstract., in particular line 30 “There was a significant reduction (56.95%) in total antibiotic usage”. Based on table 2, I suggest the change is more nuanced than this, particularly without the amoxicillin–clavulanic acid finding.

Lines 34-6: The statement reporting the cause of the changes observed is not supported by the methods or data presented (E.g. “Such decline …is explained by high awareness and antimicrobial stewardship programs supporting optimal and appropriate use of antibiotics in these hospitals”. Please consider removing it/ rewriting it.

It would be helpful to know in the abstract that the study population consisted of cats and dogs.

Line 27: “top five antibiotics”: What outcome does this relate to?

Line 37: “Furthermore, concerns were raised about the relatively high usage of third-generation cephalosporins”: Who raised these concerns?

Introduction

The authors provide a succinct overview of the background to the study. Further context would be helpful, particularly for international readers. For example: how prevalent is having pet cats or dogs in Thailand? How common is seeking care from veterinarians for cats and dogs, particularly when antibiotics are available over the counter? Are most treatment fees for companion animals paid out of pocket or are pets insured? Additional details about the veterinary sector would also be helpful: for example, are most communities served by veterinary teaching hospitals; does their case-mix differ from non-teaching hospitals/clinics (e.g. inpatient vs outpatient, severity of condition?). It would also be helpful to know that antibiotic usage guidelines for cats and dogs introduced in 2016

Line 56: Any antibiotic use – excessive or not – drives AMR. Please consider rephrasing this sentence.

Methods:

The study objective given in the methods (line 97) differs from that reported in the abstract (line 22). The reported methods are insufficient to reliably “identify trends over time” for several reasons.

Firstly, how were the longitudinal analyses undertaken and how was the study length (3 years) decided upon? These details are currently missing from the methods. The study period includes the introduction of antibiotic usage guidelines for cats and dogs in 2016 (line 260). Is having one single year of data pre and post their introduction (two data points) sufficient to support some of the claims made?

Secondly, I have some concerns about the sampling: Hospitals D and G did not supply any data in 2015, but they did in 2016 and 2017 (table1): this may have introduced bias as their prescribing practices could differ from the other hospitals in the study.

Similarly, the proportion of the sample that were cats increased from 19.5% in 2015 to 22.0% in 2017. How confident are the authors that the changes observed in antibiotic use are not due to changes in case mix? I suggest undertaking sub-group analysis for cats and dogs and presenting their results separately. Antibiotic use is known to differ between these species, for example the use of injectable antibiotics in cats who can be difficult to administer tablets to.

The study excluded animals attending for vaccination. Did the study include inpatient and outpatients? Did each hospital provide similar services (primary, secondary, tertiary care?).

More information is needed about when and how the hospitals were approached about being in the study (i.e. with respect to data collection).

Further details are also warranted about the antibiotic usage data: was this routine data collected as part of the companion animals’ care? Is it prescribing data or usage data? Was it recorded electronically or on paper records? Line 227 mentions electronic records whilst line 242 states “there is no electronic prescription database in all these hospitals”. Who collected the data needed for the study? Was this done electronically or on paper records? Was the data extracted at the event, animal, owner or hospital level?

Managing the data of over 1 million animals seeking treatment in excel would be challenging: What steps were taken to prevent and monitor accidental overwriting of cells?

I suggest moving lines 129-132 to the Ethical Approval Section

Results

To aid assessment of the generalisability of the sample, please could the authors consider providing a comparison of the characteristics if the hospitals that did/ did not agree to participate.

I suggest presenting the data in table 2 also in graphical form (perhaps as line chart) to enable easier comparison of the changes in usage of each antibiotic over the study period.

Line “Commonly prescribed”: Does this mean the antibiotic with the most prescribing events? Or the highest total weight prescribed? Please reword to make clearer throughout this paragraph. Similarly, the use of “top 5” needs a clearer definition.

Please standardise the use of decimal places.

Please add percentages to the tables.

Discussion

Some interesting ideas are presented in the discussion. It would, however, benefit from a clearer structure. For example, the addition of specific sections discussing the strengths and limitations of the study. These include the age of the data, that the conditions prompting antibiotic use and whether this was in line with best practice was guidelines was not studied. Furthermore, an average weight based on a Dutch population was applied to a Thai population.

Please revisit the proposed causes behind the observed fall in CIA use in the paragraph beginning line 256 to better reflect the methodological limitations. How confident are the authors that 2015’s result was not an outlier, and the subsequent fall was not regression to the mean?

I would also encourage the authors to develop their conceptual model of what drives antibiotic use beyond a lack of

awareness or guidelines in the discussion. There are many social, economic, cultural, historical factors. Perhaps these references might help:

Denyer Willis L, Chandler C. Quick fix for care, productivity, hygiene and inequality: reframing the entrenched problem of antibiotic overuse. BMJ Glob Health. 2019 Aug 15;4(4):e001590. doi: 10.1136/bmjgh-2019-001590

Tompson AC, Manderson L, Chandler CIR. Understanding antibiotic use: practices, structures and networks. JAC Antimicrob Resist. 2021 Oct 4;3(4):dlab150. doi: 10.1093/jacamr/dlab150

Line 233: please delete the extra the

6. PLOS authors have the option to publish the peer review history of their article (what does this mean? ). If published, this will include your full peer review and any attached files.

**Do you want your identity to be public for this peer review?** For information about this choice, including consent withdrawal, please see our Privacy Policy .

Reviewer #1: No

Reviewer #2: No

---

## [Author Response · Author response to Decision Letter 1]

1 May 2025

Significant changes

To facilitate the review process, we have included two versions of the manuscript: an annotated version highlighting the changes and a clean version for straightforward reading. The significant changes made in this revision are as follows:

- Table 2 has been renumbered as Supplementary Table 1 and modified to become Figure 1.

- Table 3 has been revised to correct decimal places and recheck the numerical results. This table has been renumbered as Table 2 in the revised version.

- Figure 1 has been renumbered as Figure 2 in the revised version.

- Figure 2 has been renumbered as Figure 3 in the revised version.

- Figure 3 has been renumbered as Figure 4 in the revised version.

Point-by-point response to editor and reviewers’ comments

Academic Editor (Journal Requirements)

Response: We have followed PLOS ONE’s style requirements in our manuscript.

“This study was funded by the Food and Agriculture Organization of the United Nations (OSRO/RAS/502/USA) and implemented by the International Health Policy Program Foundation.”

Response: We have added the Role of Funder statement in our cover letter: “This study was funded by the Food and Agriculture Organization of the United Nations (OSRO/RAS/502/USA) and implemented by the International Health Policy Program Foundation. The funders had no role in study design, data collection and analysis, decision to publish, or preparation of the manuscript.” We also deleted Funder and role from the Acknowledgements and Funding sections of our manuscript.

“We thank the veterinary teaching hospitals for providing information on antibiotic use and our colleagues at Mahidol University for assisting with data cleansing. We also acknowledge the Food and Agriculture Organization of the United Nations and the International Health Policy Foundation, Thailand, for their funding support and valuable suggestions.”

“This study was funded by the Food and Agriculture Organization of the United Nations (OSRO/RAS/502/USA) and implemented by the International Health Policy Program Foundation.”

Response: Thank you for your feedback regarding the Acknowledgments and Funding Statement. We have removed the funding information from the Acknowledgments section as requested. We confirm that the existing Funding Statement is correct as it stands. We have noted this in our cover letter.

Reviewer #1:

1. I suggest to revise the manuscript for the English because in some portion it is difficult do understand.

Response: We have resubmitted our manuscript to a professional English language editing service to enhance the clarity and quality of our writing. Please find the additional certificate for English proofreading in the system.

2. I also suggest to verify the references in the text because some are not perfectly connected to the sentence and there are some newer references you can add, for example, for the antimicrobial use in veterinary hospitals

Response: We have carefully checked all references and verified that they are accurately cited and relevant to the manuscript content. Additional updated references concerning antibiotic use in the veterinary teaching hospital have been incorporated into the Introduction section.

“Veterinary teaching hospitals in Thailand share many of the same core functions and characteristics as those in other countries, but they also reflect the unique context of Thai veterinary education, animal health needs, and cultural attitudes toward animals [11-13].”

3. I would suggest to expired the antimicrobials in table 2 in grams instead of mg

Response: We changed the unit from mg to grams. Besides, as per the related comment of Reviewer #2, we changed this table into a Figure, now Fig. 1, and we moved the numerical details to Supplementary Materials 2.

Reviewer #2:

1. Antibiotic usage in companion animals is little researched compared to livestock, particularly outside of high-income countries. This manuscript therefore provides novel insight by investigating the nature and type of antibiotics used to treat cats and dogs in Thailand. To do this, the authors descriptively analyse data collected from veterinary teaching hospitals. Whilst this is an important knowledge gap, I have concerns about the methods used and the claims made about longitudinal trends.

Response: We gratefully thank the reviewer for the valuable, insightful comments. We have now revised the manuscript according to the reviewer’s comments, focusing especially on the method section.

Abstract

2. I suggest the authors reconsider the results they present in the abstract., in particular line 30 “There was a significant reduction (56.95%) in total antibiotic usage”. Based on table 2, I suggest the change is more nuanced than this, particularly without the amoxicillin–clavulanic acid finding.

Response: We have revised the sentence to provide clearer from “There was a significant reduction (56.95%) in total antibiotic usage (mg) from 2015 to 2017” to “From 2015 to 2017, total antibiotic usage (kg) decreased significantly (i.e., by 57.0%), with a particularly notable reduction of 78.2% observed for amoxicillin–clavulanic acid.”

3. Lines 34-6: The statement reporting the cause of the changes observed is not supported by the methods or data presented (E.g. “Such decline …is explained by high awareness and antimicrobial stewardship programs supporting optimal and appropriate use of antibiotics in these hospitals”. Please consider removing it/ rewriting it.

Response: We removed the sentence as suggested.

4. It would be helpful to know in the abstract that the study population consisted of cats and dogs.

Response: We included the number of dogs and cats involved in the study, as suggested. “In total, we included 938,522 dogs and 242,342 cats in our study.”

5. Line 27: “top five antibiotics”: What outcome does this relate to?

Response: These top five antibiotics account for over 90% of total antibiotic use in our study over the three-year period.

6. Line 37: “Furthermore, concerns were raised about the relatively high usage of third-generation cephalosporins”: Who raised these concerns?

Response: We highlighted this point ourselves, as the upward trend in drug usage over the three years is concerning—particularly because the drugs involved are from higher generations.

We have revised the sentence in the abstract as follows:

“Furthermore, we observed a relatively high usage of third-generation cephalosporins, which may contribute to the emergence of antimicrobial resistance in companion animals.”

Introduction

7. The authors provide a succinct overview of the background to the study. Further context would be helpful, particularly for international readers. For example: how prevalent is having pet cats or dogs in Thailand?

Response: According to a previous study, the estimated number of owned dogs in Thailand was approximately 11.2 million [9]. However, data on the cat population were not available. This limitation is mainly due to the absence of an effective pet registration system in the country, making it difficult to obtain accurate figures for both dogs and cats. We added, “Dogs are the most common pets raised in Thailand. The human-to-owned dog ratio in Thailand was estimated at 6.4, based on an owned dog population of approximately 11.2 million [9].”

8. How common is seeking care from veterinarians for cats and dogs, particularly when antibiotics are available over the counter?

Response: While precise figures on veterinary care-seeking for cats and dogs in our country are unavailable, it is generally understood that accessing veterinary services can vary based on factors like pet owner income, location (urban vs. rural), and awareness. Regarding antibiotics, it is crucial to clarify that over-the-counter sales of antibiotics are illegal in Thailand for both human and animal use. Prescriptions are required and can only be issued by licensed professionals.

9. Are most treatment fees for companion animals paid out of pocket or are pets insured?

Response: Pet insurance penetration in Thailand is very low. Consequently, the vast majority of companion animal treatment fees are paid out-of-pocket by pet owners.

10. Additional details about the veterinary sector would also be helpful: for example, are most communities served by veterinary teaching hospitals; does their case-mix differ from non-teaching hospitals/clinics (e.g. inpatient vs outpatient, severity of condition?). It would also be helpful to know that antibiotic usage guidelines for cats and dogs introduced in 2016

Response: Veterinary teaching hospitals in Thailand serve as crucial training centers for undergraduate and postgraduate veterinary students, and they handle a diverse case mix. They manage both primary care and complex cases, similar to teaching hospitals in other countries. However, they are also integrated within the unique context of Thai veterinary education. While specific data on case-mix differences between teaching and non-teaching hospitals/clinics (e.g., inpatient vs. outpatient, severity of condition) is limited, teaching hospitals typically see a higher proportion of complicated and referral cases due to their specialized resources and expertise.

Regarding antibiotic usage guidelines, the Veterinary Council of Thailand issued guidelines for cats and dogs in 2017, following the launch of the National Strategic Plan to reduce antimicrobial use in 2016. Furthermore, it is worth noting that the Thai veterinary community has been discussing and promoting continuing education on antimicrobial stewardship in small animals since 1997, indicating a long-standing awareness of this critical issue.

11. Line 56: Any antibiotic use – excessive or not – drives AMR. Please consider rephrasing this sentence.

Response: We have rephrased the sentence from “Antibiotics have been excessively used in humans and animals, resulting in resistance to pathogens and other commensal bacteria” to “Widespread and frequent use of antibiotics in humans and animals has contributed to antimicrobial resistance among both pathogens and commensal bacteria.”

Methods:

12. The study objective given in the methods (line 97) differs from that reported in the abstract (line 22). The reported methods are insufficient to reliably “identify trends over time” for several reasons.

Response: We have changed “identify trends over time” to “identify patterns” for our objectives in the Materials and Methods part in Lines 110

13. Firstly, how were the longitudinal analyses undertaken and how was the study length (3 years) decided upon? These details are currently missing from the methods. The study period includes the introduction of antibiotic usage guidelines for cats and dogs in 2016 (line 260). Is having one single year of data pre and post their introduction (two data points) sufficient to support some of the claims made?

Response: Regarding the study period (2015-2017) in relation to antibiotic stewardship initiatives. We wish to clarify the timeline: the national strategic plan on antimicrobial resistance (AMR) was endorsed in 2016 (for the implementation period 2017–2021/22), and additionally, specific antibiotic usage guidelines for small animals were launched by the Veterinary Council of Thailand in 2017.

Our study period therefore covers:

- 2015: Baseline year before both the national plan endorsement and the specific veterinary guidelines.

- 2016: Year the national plan was endorsed, but before its official implementation period and before the specific veterinary guidelines.

- 2017: The first year of the national plan's implementation and the year the specific small animal antibiotic guidelines were launched by the Veterinary Council of Thailand.

Our study aimed to capture the antibiotic usage landscape in small animal practice in Thailand during this critical transition period. While 2015 represents a pre-intervention baseline and 2017 represents the initial year where both the national strategy's implementation began and specific veterinary guidelines were introduced, the primary goal was not solely to measure the immediate, causal impact of these combined initiatives with these data points alone. Rather, the study was designed as part of the broader national effort to understand baseline antibiotic use across sectors, spurred by the strategic plan, as national focus on AMR intensified and specific professional guidance emerged.

We acknowledge that assessing the long-term effectiveness and direct impact of these combined initiatives would ideally require a longer time series with more data points post-2017. We will refine the Discussion section to:

“Second, the interpretation of our findings should also consider the concurrent introduction of major antimicrobial stewardship initiatives. For example, Thailand’s National Strategic Plan on AMR was launched in 2016 [28], and specific Veterinary Council guidelines for companion animal antibiotic use followed in 2017 . Our study, which covers the period 2015–2017 captures this pivotal transition, and therefore provides a valuable baseline that reflects the lead-up to as well as the initial implementation phase. We recognize explicitly that having only one full year pre- (2015) and one year coinciding with the launch/start of these initiatives (2017) limits our ability to definitively assess their sustained impact. However, these data effectively highlight usage patterns within veterinary teaching hospitals right at the point these crucial guidelines and strategies were introduced.”

14. Secondly, We have some concerns about the sampling: Hospitals D and G did not supply any data in 2015, but they did in 2016 and 2017 (table1): this may have introduced bias as their prescribing practices could differ from the other hospitals in the study.

Response: Thank you for your kind consideration. We have already added this to the limitation in our discussion section.

“(6) One limitation of our study was the absence of antibiotic usage data from two hospitals in 2015. However, we addressed this by estimating the missing values using the median antibiotic usage figures of these hospitals in 2016 and 2017 [44], reflecting a plausible usage trend approach. A subsequent sensitivity analysis showed that total antibiotic usage, after imputation, differed by only 1.18% from the original dataset, excluding the missing 2015 data (Table S3). This minimal difference suggests that the available data remain a valid and representative estimate of the overall antibiotic usage across all eight hospitals over the 2015–2017 period.”

15. Similarly, the proportion of the sample that were cats increased from 19.5% in 2015 to 22.0% in 2017. How conf

---

## [Decision Letter · Decision Letter 1]

3 Jul 2025

PONE-D-24-54379R1Antibiotic use in companion animals in veterinary teaching hospitals in ThailandPLOS ONE

Dear Dr. Sakcamduang,

Thank you for submitting your manuscript to PLOS ONE. After careful consideration, we feel that it has merit but does not fully meet PLOS ONE’s publication criteria as it currently stands. Therefore, we invite you to submit a revised version of the manuscript that addresses the points raised during the review process.

We look forward to receiving your revised manuscript.

Kind regards,

Md. Tanvir Rahman, DVM, MSc (Canada), PhD (UK), FBAS

Academic Editor

PLOS ONE

Journal Requirements:

Additional Editor Comments:

i recommend minor revision asv suggested by one reviewer on the submitted revised version.

Reviewers' comments:

Reviewer's Responses to Questions

**Comments to the Author**

1. If the authors have adequately addressed your comments raised in a previous round of review and you feel that this manuscript is now acceptable for publication, you may indicate that here to bypass the “Comments to the Author” section, enter your conflict of interest statement in the “Confidential to Editor” section, and submit your "Accept" recommendation.

Reviewer #2: (No Response)

2. Is the manuscript technically sound, and do the data support the conclusions?

Reviewer #2: Partly

3. Has the statistical analysis been performed appropriately and rigorously? 

Reviewer #2: No

4. Have the authors made all data underlying the findings in their manuscript fully available?

Reviewer #2: Yes

5. Is the manuscript presented in an intelligible fashion and written in standard English?

Reviewer #2: Yes

6. Review Comments to the Author

Reviewer #2: Methods: A number of analytical steps are not reported in the methods. Particularly, i) how the trends in antibiotic use were calculated, ii) the definition of the ‘top five’ antibiotics and iii) the new sensitivity analysis used to explore the impact of only having six clinics at baseline. Please ensure that methods are reported for all the results presented and that this description is located within the methods section, eg line 179 “We also tracked antibiotic usage from 2015 to 2017 and classified the proportion that were 180 listed on the WHO List of CIA for Human Medicine (version 2019)”.

Methods: The manuscript describes the change in antibiotic use. For example, “decreased significantly” (line 34) and “dramatically decreased” (line 201). The manuscript makes much of these observed changes and therefore the study would be greatly strengthened if these changes were evaluated using statistical significance testing.

Results: I am unable to access the supplementary files. Regardless, it would be helpful if the total weight of antibiotics used by year was presented in the main text (line 173).

Methods: I am unqualified to comment on the appropriateness of the newly added “plausible usage trend approach” used to evaluate the impact of including only 6 clinics at baseline.

7. PLOS authors have the option to publish the peer review history of their article (what does this mean? ). If published, this will include your full peer review and any attached files.

**Do you want your identity to be public for this peer review?** For information about this choice, including consent withdrawal, please see our Privacy Policy .

Reviewer #2: No

---

## [Author Response · Author response to Decision Letter 2]

4 Aug 2025

Reviewer #2:

Methods: A number of analytical steps are not reported in the methods. Particularly,

i) how the trends in antibiotic use were calculated,

Response: Thank you for your comment regarding how the trends in antibiotic use were calculated. We would like to clarify that our study did not perform statistical analyses to evaluate changes over time. Instead, the graphical presentations in our manuscript reflect descriptive analysis, illustrating the observed patterns and volumes of antibiotic consumption across the study years (2015 to 2017). Specifically, the patterns shown in the figures were derived from calculated quantities of antibiotics used per kilogram of animal weight per year, aggregated by antibiotic class and year, based on prescription and usage records from the veterinary teaching hospitals. These aggregated data were then plotted to visually demonstrate changes or consistencies in antibiotic usage over the three-year period.

We added “descriptive statistics” to confirm the methods used for our analysis: Data entry and descriptive analysis were performed using Microsoft Excel (version 2016).

To clarify and improve scientific accuracy, we have revised the manuscript's wording to eliminate any potential misinterpretation regarding statistical analysis. Please refer to the manuscript version highlighting the revisions.

ii) the definition of the ‘top five’ antibiotics,

Response: Thank you for your valuable comment regarding the definition of the “top five” antibiotics in our manuscript. We would like to clarify that the “top five” antibiotics refer to those with the highest total usage calculated in 1) total grams, and 2) milligrams of active ingredient per kilogram of animal body weight per day (mg/kg/day) within the study period.

Therefore, we have enhanced the Materials and Methods to include the following information: "The top five antibiotics have been identified based on their ranking in terms of total grams of antibiotic usage, as well as the measurement of antibiotic consumption expressed in milligrams per kilogram per year."

Furthermore, we have clarified the term "top five" in the text to specify that it refers to the highest rankings in terms of both total grams and milligrams per kilogram per day of antibiotic use.

iii) the new sensitivity analysis used to explore the impact of only having six clinics at baseline.

Response: The sensitivity analysis showed that the overall patterns and key findings remained similar, indicating that the results were robust and not significantly influenced by the change in the number of clinics included at baseline. Full details of the sensitivity analysis have been added to the Materials and Methods of the revised manuscript as follows:

“In 2015, we encountered a lack of data on antibiotic use from two out of eight veterinary teaching hospitals. To address this gap, we estimated the total antibiotic use for these two hospitals by using the median value of antibiotic consumption recorded in 2016 and 2017. Following this, we calculated the overall antibiotic usage from 2015 to 2017. To evaluate our estimates, we conducted a sensitivity analysis comparing the existing data on total antibiotic use across 2015, 2016, and 2017 with our estimated data for the two hospitals that were missing information in 2015. This analysis aimed to determine the difference in total antibiotic use between the two estimation methods and assess whether we could rely on the existing data without the estimated information from those two hospitals.”

Please ensure that methods are reported for all the results presented and that this description is located within the methods section, eg line 179 “We also tracked antibiotic usage from 2015 to 2017 and classified the proportion that were 180 listed on the WHO List of CIA for Human Medicine (version 2019)”. Response: We have carefully reviewed the manuscript to ensure that all methodological descriptions related to the results are clearly detailed and located within the Materials and Methods. In particular, the procedure for tracking antibiotic usage from 2015 to 2017 and classifying the proportion of antibiotics listed on the WHO List of Critically Important Antimicrobials (CIA) for Human Medicine (version 2019) has been explicitly described in the Methods.

We have added clarifications on how antibiotics were identified, categorized according to the WHO CIA list, and how their usage proportions were calculated for each year. This ensures transparency of the analyses reported, including the classification and quantification of antibiotic consumption.

Methods: The manuscript describes the change in antibiotic use. For example, “decreased significantly” (line 34) and “dramatically decreased” (line 201). The manuscript makes much of these observed changes and therefore the study would be greatly strengthened if these changes were evaluated using statistical significance testing. Response: We would like to clarify that our study was primarily descriptive in nature and aimed to provide an overview of antibiotic consumption patterns in companion animals over the study period. Due to the retrospective design and the nature of the data collected—aggregated antibiotic usage from veterinary teaching hospitals with varied reporting and relatively small sample size for formal statistical trend analysis—we were unable to perform robust statistical tests to assess the significance of changes over time.

Additionally, the dataset’s structure, consisting of aggregated mg/kg/year antibiotic usage without repeated individual-level measures or consistent sampling units across clinics in some years, limits the application of conventional inferential statistical methods for trend analysis. The graphs and qualitative descriptions were intended to highlight observed patterns and potential shifts in antibiotic use rather than to confirm statistically significant changes.

Therefore, to reduce potential confusion in the manuscript, we revised the wording in several sections to avoid implying that statistical analyses were performed, as you asked. These changes

aim to enhance clarity and prevent any possible misinterpretation by readers, as we discussed in the first question.

Results: I am unable to access the supplementary files. Regardless, it would be helpful if the total weight of antibiotics used by year was presented in the main text (line 173).

Response: In response to your valuable suggestion, we have already uploaded the revised supplementary files and included the total weight of antibiotics used by year (2015 to 2017) directly into the main text of the revised manuscript. This inclusion provides a clear and immediate presentation of the overall antibiotic quantities used across the study period, thereby improving the accessibility and clarity of key results.

Methods: I am unqualified to comment on the appropriateness of the newly added “plausible usage trend approach” used to evaluate the impact of including only 6 clinics at baseline.

Response: We recognize the reviewer’s concern about the baseline data being derived from only six clinics in 2015. To address this limitation, we performed a sensitivity analysis using estimated values based on the median from 2016 to 2017. This analysis was conducted to evaluate how the absence of data could influence the overall estimates of antibiotic usage. Using the median to impute missing values is a widely accepted and practical method in descriptive studies, particularly when the aim is to minimize the effects of outliers (Zainal et al., 2023).

In our study, we used this method to estimate the 2015 data for the two hospitals with missing records. We compared the total antibiotic usage using two methods: (1) the actual data from the six hospitals, and (2) the augmented dataset that included the median-imputed values for the two hospitals. This comparison revealed only a 1.18% difference between the two totals, indicating that the imputed data had a minimal impact on the overall usage estimate.

Based on this outcome, we concluded that it was reasonable to proceed with our analysis using the original dataset without imputation. We have clarified this methodological decision in the revised manuscript to enhance transparency and address any potential concerns regarding data completeness and representativeness.

It is important to highlight that these six hospitals represent the largest tertiary veterinary teaching hospitals in Thailand, covering 8 out of the 13 total vet teaching hospitals nationwide.

Furthermore, our antibiotic usage estimates are expressed in mg/kg/day, which involves normalization by the number and weight of treated cases. This normalization accounts for differences in clinic size and case load, allowing the mg/kg/day estimates to provide a meaningful reference point for comparing antibiotic use patterns, even when data are limited to certain clinics or years.

Thus, while formal statistical testing was not applied, we believe that this approach and normalization method provide a reasonable and informative framework for interpreting patterns in antibiotic usage within this specific context.

Reference

Zainal H, Wan Yusoff WS, Ab Latif WAW, Mohd Noor N, Ahmad S, Ismail AI, et al. (2023) Handling missing data: A practical guide for researchers. Data Brief 47: 100341. https://doi.org/10.1016/j.dajour.2023.100341

---

## [Editor Report · Decision Letter 2]

6 Aug 2025

Antibiotic use in companion animals in veterinary teaching hospitals in Thailand

PONE-D-24-54379R2

Dear Authors,

We’re pleased to inform you that your manuscript has been judged scientifically suitable for publication and will be formally accepted for publication once it meets all outstanding technical requirements.

Kind regards,

Md. Tanvir Rahman, DVM, MSc (Canada), PhD (UK), FBAS

Academic Editor

PLOS ONE

Additional Editor Comments (optional):

Thanks for addressing the comments of the reviewers.
---

## [Editor Report · Acceptance letter]

PONE-D-24-54379R2

PLOS ONE

Dear Dr. Sakcamduang,

I'm pleased to inform you that your manuscript has been deemed suitable for publication in PLOS ONE. Congratulations! Your manuscript is now being handed over to our production team.

Kind regards,

on behalf of

Professor Md. Tanvir Rahman

Academic Editor

PLOS ONE